# Mitigation of Adversarial Policy Imitation via Constrained Randomization of Policy (CRoP)

**Nancirose Piazza,**[1] **Vahid Behzadan**[1]

SAIL Lab[1]
University of New Haven
West Haven, CT, USA
npiaz1@unh.newhaven.edu
vbehzadan@unh.newhaven.edu

## Abstract

Deep Reinforcement Learning (DRL) policies are vulnerable to unauthorized replication attacks, where an adversary exploits imitation learning to reproduce target policies from observed behavior. In this paper, we propose Constrained Randomization of Policy (CRoP) as a mitigation technique against such attacks. CRoP induces the execution of sub-optimal actions at random under performance loss constraints. We present a parametric analysis of CRoP, address the optimality of CRoP, and establish theoretical bounds on the adversarial budget and the expectation of loss. Furthermore, we report the experimental evaluation of CRoP in Atari environments under adversarial imitation, which demonstrate the efficacy and feasibility of our proposed method against policy replication attacks.

## Introduction

Deep Reinforcement Learning (DRL) is a learning framework for sequential decision-making leveraging neural networks for generalization and function approximation. With the growing interest in DRL and its integration in commercial and critical systems, the security of such algorithms have become of paramount importance (Behzadan and Munir 2018).

In tandem with DRL, similar advancements have been made in Imitation Learning (IL) techniques that utilize expert demonstrations to learn and replicate the expert's behavior in sequential decision making tasks. Deep Q-Learning from Demonstration (DQfD)(Hester et al. 2017) is an IL variant that has enabled DRL agents to converge quicker to an optimal policy. However, recent work in (Behzadan and Hsu 2019a) and (Chen et al. 2020) demonstrate that IL can also be exploited by adversaries to replicate other agents' policies from passive observation of their behavior. This gives rise to risks concerning intellectual property and adversarial information gain for more effective active attacks. Current state of the art in countering such attacks include watermarking (Behzadan and Hsu 2019b)(Chen et al. 2021), which enables the post-attack identification of replicated policies.

In this paper, we propose an active mitigation technique against policy imitation attacks, named Constrained Randomization of Policy (CRoP). The proposed technique is based on intermittent randomization of a trained policy, constrained

on a threshold for maximum amount of acceptable loss in the expected return. The goal is to increase the adversary's imitation training cost, measured as the minimum number of training iterations and observed demonstrations required for training a replica that matches the target policy's performance.

The main contributions of this paper are: (1) We propose and formulate CRoP as a mitigation technique against adversarial policy imitation, (2) We present a formal analysis of the bounds on expected loss of optimality under CRoP, (3) We formally establish bounds on the adversary's imitation cost induced by CRoP. (3) We report the results of empirical evauation of adversarial imitation via DQfD against CRoP agents in classical DRL benchmarks, and demonstrate the efficacy and feasibility of CRoP in those settings.

The remainder of this paper is organized as follows: we introduce CRoP as a mitigation technique against policy imitation, and analyze the optimality of a CRoP policy in relation to an optimal policy. The analysis further establishes a lower bound on adversarial budget induced by CRoP. The following section reports the experimental evaluation of CRoP in three Atari benchmark environments, along with measurements of the training and test-time performance of DQfD-based adversarial imitation learning agents targeting CRoP-enabled policies. We conclude the paper with a summary of findings and remarks on future directions of research.

## Constrained Randomization of Policy

In the remainder of this paper, we assume the target policy aims to solve a Markov Decision Process (MDP) denoted by the tuple $< S, A, R, T, \gamma >$ where $S$ is a finite state space, $A$ is a finite action space, $T$ defines the environment's transition probabilities, a discount value $\gamma \in [0, 1)$, and a reward function $R : S \times A \rightarrow [0, 1]$. The solution to this MDP is a policy $\pi : S \rightarrow A$ that maps states to actions. An agent implementing a policy $\pi$ can measure the value of a state $V(s) = \max_a (r_{s,a} + \gamma V(s'))$, where $s'$ is the next state. Similarly, the value of a state-action pair is given by $Q(s, a) = \max_a (r_{s,a} + \gamma Q(s', a'))$ where $s'$ is the next state and $a'$ is the next action.

Constrained Randomization of Policy (CRoP) is an action diversion strategy from an optimal policy under constrained performance deviation from optimal. Let $\hat{a} \in \hat{A}$ where $\hat{a}$ are

candidate actions that satisfy $|Q(s, \pi(s)) - Q(s, \hat{a}_i)| < \rho$. In other words, $\hat{A}$ is the space of all candidate actions for $s \in S$ excluding the optimal action $\pi(s)$. We define CRoP as the function below:

$$f(s) = \begin{cases} \pi(s) & Pr(\delta) \text{ or } \hat{A} = \emptyset \\ \hat{A}_{\hat{a} \sim U(\hat{A})} & Pr(1-\delta) \end{cases} \quad (1)$$

Where $U(\hat{A})$ is the uniform distribution over $\hat{A}$. This definition of $\rho$ threshold is the difference of Q-values. We have three variations of $\rho$ for CRoP: Q-value difference (Q-diff) as described in Equation 1, and two measures inspired by the advantage function: advantage-inspired difference (A-diff), and positive advantage-inspired difference (A$^+$-diff). A-diff CRoP is thus defined as:

$$\tilde{A}(s_t, a_t) = Q(s_t, a_t) - V(s_{t-1}) > -\rho \quad (2)$$

A$^+$-diff's $\rho$ has the condition $\hat{A}(s_t, a_t) \geq 0$. A-diff and A$^+$-diff's $\rho$ are interpreted as 1-step hindsight estimation which is relevant to the trajectory taken instead of only pure future estimate as with Q-diff, eg. played badly, now play safe vs. plan to feint ahead. However, the selection of $\rho$ should consider estimation error due to either finite training or function approximation. One can look to the analysis of learning complexity as a method of finding error bounds to derive a safety margin for $\rho$. We choose these three threshold variations because their performance vary across the environments, implying that being able to successfully deviate from optimal policy is conditioned on the environment dynamics itself as well as the defender's tolerance for loss which may not be captured by a single threshold such as Q-diff's. We cannot use the traditional Advantage since $V(s) = \max_a Q(s, a)$ implies Q-diff's implementation would have a similar impact in regards to threshold. Additionally, it is important to recognize that CRoP is similar to that of a $\epsilon$-greedy policy; however, the difference lies on the constraint expected loss that $\epsilon$-greedy does not guarantee.

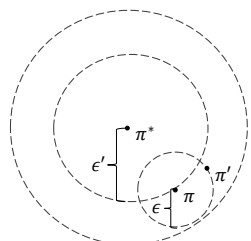

Figure 1: Visualization that $\pi'$ is an $(\epsilon + \epsilon')$-optimal to $Q^*/V^*$

By definition, a policy $\pi$ is $\epsilon$-optimal if there exists a non-negative constant $\epsilon$ such that $v^\pi(x) \geq V(x) - \epsilon$ for all initial states $x$ in $S$. Some definitions include that it occurs at $P(1-\delta)$, which we will abide by. In other words, $\epsilon$-optimal policies are within an $\epsilon$ neighborhood of $V^*$, specifically $V^* - V^\pi < \epsilon$ for all $a \in A$ and $s \in S$ at probability $(1-\delta)$. As illustrated in Figure 1, $\pi^*$ is the optimal and greedy policy extracted from $V^*$ where $\pi$ is the extracted policy from $V^\pi$ and $\pi'$ is the extracted policy from $V^{\pi'}$, we see that $\pi'$ may

be expressed as an $(\epsilon + \epsilon')$-optimal to $Q^*/V^*$ when evaluated for the initial states and to follow a greedy policy thereafter. Since we do not assume $\pi$ to be an optimal policy, it is possible for $\pi'$ to be more optimal than $\pi$. However, it is noteworthy that an evaluation of optimality based on the difference to the value function does not imply extracted policies with small error to $V^*$ resemble the optimal policy when assessed on behavioral differences. Theorem 1 establishes that CRoP policy $f$ is at worst $(\epsilon + \epsilon')$-optimal to $Q^*$ at probability $(1-\delta)$ as an evaluation of the initial states assuming a greedy policy is followed after. However, an evaluation for committing to following CRoP thereafter (or for any evaluation of a trajectory under CRoP via sequence of Q-values) will have compounding sub-optimality or horizon dependent error. However, $\epsilon$-optimal can have compounding error in expectation regardless of CRoP. Therefore, instead for any fix length horizon $T$, we know that $\pi'$ will be $(T \times \epsilon + \epsilon')$-optimal to the sum of the taken trajectory's states taken under $\pi'$ evaluated by $V^*$. In other words, $\sum^T V^*(s) - \sum^T V'(s) < T \times \epsilon + \epsilon'$ for a finite state trajectory set $s \in S$ of cardinality $T$ under $\pi'$; however, this is different than a trajectory taken under $\pi^*$. We cannot evaluate an optimal trajectory from local view; however, since we allow the defender to modify $\rho$, one can simply stop the deviating behavior to cease and bound the compounding error at state $s$, though the error in a trajectory measured by difference in the sum of Q-values from $Q^*$ may continue to compound.

**Theorem 1** *Given $Q^*(s_t, a_t) - Q^\pi(s_t, a_t) < \epsilon'$ at probability $(1-\delta)$ and $|Q^\pi(s_t, a_t) - Q^{\pi'}(s_t, a_t)| \leq \epsilon$ for all $s \in S$ and $a \in A$, then $Q^*(s_t, a_t) - Q^{\pi'}(s_t, a_t) \leq \epsilon + \epsilon'$ at probability $(1-\delta)$. $\pi'$ is an $(\epsilon + \epsilon')$-optimal to $Q^*/V^*$ at probability $(1-\delta)$.*

Proof. Given $Q^*(s_t, a_t) - Q^\pi(s_t, a_t) < \epsilon'$ at probability $(1-\delta)$ and $|Q^\pi(s_t, a_t) - Q^{\pi'}(s_t, a_t)| \leq \epsilon$ for all $s \in S$ and $a \in A$, then $Q^*(s_t, a_t) - Q^{\pi'}(s_t, a_t) \leq \epsilon + \epsilon'$ at probability $(1-\delta)$.

Let $Q_{diff} = Q^*(s_t, a_t) - Q^f(s_t, a_t) + |Q^f(s_t, a_t) - Q^{\pi'}(s_t, a_t)|$. Given that $Q(s, a) \in (0, \frac{1}{1-\gamma})$, at $(1-\delta)$ probability:

$$Q^*(s_t, a_t) - Q^{\pi'}(s_t, a_t) \leq Q_{diff} \leq \epsilon + \epsilon' \quad (3)$$

We consider two common approaches to IL: (1) Behavioral Clones (BC) which are supervised learners, and (2) reinforcement learning from demonstration techniques such as DQfD, which augment RL with IL. Work by (Ke et al. 2020) shows that: BC minimizes the KL divergence, Generative Adversarial Imitation Learning (GAIL) (Ho and Ermon 2016) minimize the Jensen Shannon divergence and DAgger (Ross, Gordon, and Bagnell 2011) minimizes total variance. For BC, CRoP affects the maximum likelihood in a similar manner to data poisoning attacks like label flipping (Xiao, Xiao, and Eckert 2012) or class imbalance. In regard to GAIL, the discriminator from a GAN prioritizes expert experiences so unless modified for decay when out-performed, additional penalty is given to the training policy. Furthermore, when CRoP lowers the action distribution for $a^*$ according to $\delta$ probability and increases the distribution for candidate actions, it results in smaller maximal difference for DAgger.

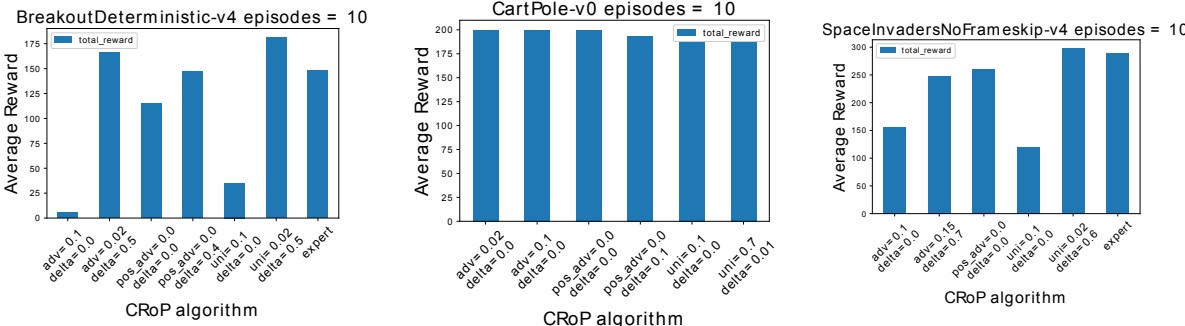

Figure 2: Test-time evaluation of target agent under various CRoP thresholds across 10 episodes

(a) Breakout      (b) Cartpole      (c) SpaceInvaders

Figure 3: Test-time evaluation of replicated policies and the target DDQN agent across 10 episodes

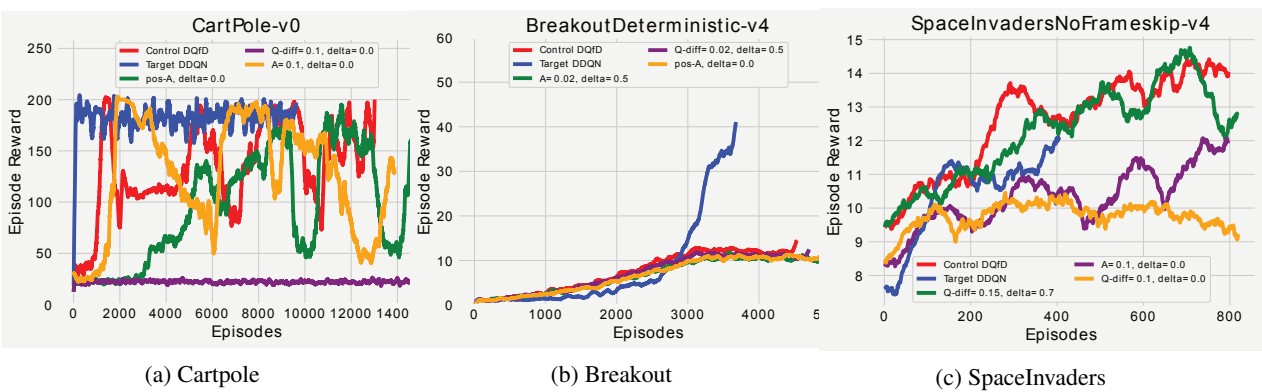

(a) Cartpole      (b) Breakout      (c) SpaceInvaders

Figure 4: Imitating DQfD agents training on CRoP-induced demonstrations

## Budget Analysis for Perfect Information Adversary

The adversary objective is one of two objectives: (1) policy replication such as behavioral imitation or structural similarity in learned parameters, (2) training computation cost reduction. A defender's objective is to minimize adversarial gain or maximize the adversary's budget while simultaneously maximizing their protected policy's return; however, since the minimization of adversarial gain results in performance deterioration, a defender must also determine the tolerance of performance deterioration.

We measure the adversary's budget in the sample quantity or trajectories that it can acquire through a passive attack. Nair and Doshi-Velez (Nair and Doshi-Velez 2020) derive upper and lower bounds on the sample complexity of direct policy learning and model-based imitation learning in relaxed problem spaces. This follows the research of RL sample efficiency and Offline RL(Levine et al. 2020). However, in this work we divert from a direct treatment of sample efficiency to consider information optimality from observed target demonstration without environment interaction. Consider the set $\mathcal{T}$ where $\tau_i$ ($\forall, \tau_i \in \mathcal{T}$) which is composed of a $T$-length chain of $(s, a)$-pairs. Assume each $(s, a)$-pair has two possible outcomes, optimal at $P(\delta)$ or sub-optimal at $P(1 - \delta)$. Assume pair and trajectory uniqueness, this would contain $2^T$ trajec-

tories where $T$ is the length of the horizon. To obtain optimal target $\pi$, we would require all trajectories except the event of a complete sub-optimal trajectory $(1 - \delta)^T$. Let $\tau_w$ be the worst-case trajectory and $\hat{m}$ be the sum of the expected number of trajectories for each sequential pull from $\mathcal{T}$. It follows that:

$$\mathbb{E}[\hat{m}] = \sum_{1}^{2^T-1} 1/(1 - P(\tau_w) + \sum_{\tau_i \in \hat{\mathcal{T}}} -P(\tau_i)) \qquad (4)$$

Intuitively, we see in the denominator the probability of pulling unseen trajectories given the trajectories in $\hat{\mathcal{T}}$ and known probability for all $\tau_i \in \hat{\mathcal{T}}$. We know an expectation on expensive to obtain informative trajectories from $\pi$. However, typically an adversary has a fixed budget and therefore we would want to know what to expect given their budget $\mathbb{B}$, here we calculate for a budget measured in optimal state-action pairs. To calculated an expected number of optimal state-action pairs, we find a $t < T$ such that:

$$\mathbb{B} \approx \sum_{i=1}^{t} \mathbb{E}[m_i] = \sum_{i=1}^{t} \frac{1}{\delta} \qquad (5)$$

Given we can reset to the previous state and resample until we obtain an optimal state-action pair. This would give an

expectation for the adversary to obtain $t$ optimal state-action pairs with $\mathbb{B}$ budget. This can be extended to the expectation of number of trajectories by approximating $\mathbb{B}$, similar to Equation 5 where we find a $t < T$, but with Equation 4.

## Policy Evaulation and Expectation of Loss

We see that the Q-value under $f$ will be either equivalent or less than the Q-value under target policy $\pi$ which dictates selected $a'$. Furthermore, the expected return $G_t^f$ for stochastic policy $f$ with uniform sampling from $\hat{A}$ is expressed as the following:

$$G_t^f = \delta \sum_{t=0,1,2...}^{N} \gamma^t \left[ r_{s_t, a_t^*} \right] + \frac{1-\delta}{|\hat{A}|} \sum_{t=0,1,2...}^{N} \gamma^t \left[ \sum_{\hat{a}_t} r_{s_t, \hat{a}_t} \right] \quad (6)$$

With Equation 6, $G_t^f$ is the weighted sum of an optimal expected return at probability $\delta$ and the expected return across all rewards given by candidate actions at probability $(1 - \delta)$. Given $G_t^*$ and $G_t^f$, the difference between the expected return in $Q$-value form is exactly:

$$G_t^* - G_t^f = (1 - \delta) \left[ Q^\pi(s_t, a_t) - \mathbb{E}[Q^f(s_t, \hat{a}_t)] \right] \quad (7)$$

Since $Q^\pi(s_t, a_t) - \mathbb{E}[Q^f(s_t, \hat{a}_t)] < \rho$, then the expectation loss $G_t^* - G_t^f \leq (1 - \delta)\rho \leq \rho$. This expectation of loss is calculated from the current state's forward estimation of future reward. We see there exists an upperbound, call it $\mathbb{E}[L]$ which is derived from the evaluation of trajectory returns taken under $f$ which is the agent's local reference. This is important to defenders because they can estimate when to cease CRoP given its local view:

$$\sum_{t=0}^{N} |Q^\pi(s_t, a_t) - \mathbb{E}[Q^f(s_t, \hat{a}_t)]| \leq N \times (1-\delta)\rho \leq N \times \rho = \mathbb{E}[L]$$
$$(8)$$

## Experimental Evaluation

We investigate DQfD as our adversarial IL method and evaluate test-time and training time performance across three Atari environments: Breakout, Cartpole, and Space Invaders. We train DQfD agents under default parameters (supplied in supplements) with CRoP induced demonstrations, a control DQfD agent (our baseline IL comparison), and a default, double DQN (DDQN) agent which provided the expert demonstrations and as well as be a baseline performance comparison for IL agents. The results of a parameter search on trained DDQN policies from Stable-Baseline Zoo (Raffin 2018) and the count of candidate timesteps are presented in the longer extension of this paper. The test evaluations of the target policy under selected thresholds of CRoP follow in Figure 2 to show that the selected $\rho$ and $\delta$ values reflect the evaluation of average reward performance. The trade-off on $\delta$ and $\rho$ is similar to an allowance of high or low variance in Q-value. One can draw a similarity of the defender's selection of $\delta$ and $\rho$ as risk-adverse, risk-neutral, and risk-seeking behaviors determined by the defender. The results, illustrated in Figure 4, demonstrate that the performance of imitated

policies generally remain below their control/baseline DQfD agents for earlier spans of training episodes. We compare it to the baseline DQfD because the baseline demonstrates the performance of a DQfD agent with no mitigation against adversarial ease-dropping on state-actions performed by the target agent. CRoP may induce variance similar to optimistic initialization, for example, work by (Kamiura and Sano 2017) and (Szita and Lörincz 2009). However, we argue that by adding deviating behavior to the assumed optimal target policy, the target policy is withholding the maximal information gain an adversary can observe, thus increasing their adversarial budget. Figure 3 depicts the comparison of test-time performance among agents trained with various values of $\delta$ and $\rho$. We emphasize the constrains in CRoP are expected loss which are not true performance loss. This supports the need for several threshold variations because policies under CRoP behaved differently across the environments.

## Conclusion

This study investigated the threat emanating from passive policy replication attacks through adversarial usage of Imitation Learning. We proposed Constrained Randomization of Policy (CRoP), a deviation from optimal policy under a threshold constraint, as a mitigation technique against such attacks. We perform a parameter search and empirically evaluate the target policy under CRoP in comparison to the target policy without protection. We analyzed its performance with regards to $\epsilon$-optimality, estimated impact on adversarial cost, and the expectation of loss. Furthermore, we empirically evaluated CRoP across 3 Atari game benchmarks, and verified the efficacy and efficiency of CRoP against DQfD-based policy replication attacks, demonstrating that it is possible for the target policy to accomplish its task while deviating behavior in a bounded manner to increase the adversarial cost for successful policy replication.

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
