# OpenReview forum: "Mitigation of Adversarial Policy Imitation via Constrained Randomization of Policy (CRoP)"
_AAAI.org/2022/Workshop/AdvML — AAAI-22 AdvML Workshop ShortPaper_

### Official Review · Reviewer_p6YY · 2021-11-24
**A good work for improving the robustness against unauthorized replication attacks**

**Rating:** 6
**Confidence:** 4

**Review:**

This work proposes Constrained Randomization of Policy (CRoP) as a deep reinforcement learning defence method against unauthorized replication attacks. Though the theoretical part needs to be organized better, the extensive experiments show the effectiveness of CRoP.

Some suggestions:

1. It's better and more clear to first define the attack objective as well as the defence objective.

2. Some symbols can be more concise, e.g. in eq(1), $\!\exists \hat{a}\in\hat{A}$ may be replaced with $\hat{A} = \emptyset$.

3. It's better to define the meaning of the symbol before it's first used, e.g. $m_n$.

4. Some theoretical results need more careful discussion, e.g. eq(9) provides an upper bound, what's the connection with defence?

---

### Decision · Program_Chairs · 2021-12-01

**Decision:**

Accept (Short Paper)

**Comment:**

The reviewer agrees to accept this paper. Please address the reviewer's comment in camera-ready. Please format your paper within 4 pages (excluding references) also.